# A Numerical Algorithm for Solving Nonlocal Nonlinear Stochastic Delayed Systems with Variable-Order Fractional Brownian Noise

Behrouz Parsa Moghaddam [1], Maryam Pishbin [1], Zeinab Salamat Mostaghim [1], Olaniyi Samuel Iyiola [2], Alexandra Galhano [3,*] and António M. Lopes [4]

1   Department of Mathematics, Lahijan Branch, Islamic Azad University, Lahijan 1477893855, Iran
2   Department of Mathematics, Clarkson University, Potsdam, NY 13699, USA
3   Faculdade de Ciências Naturais, Engenharias e Tecnologias, Universidade Lusófona do Porto, Rua Augusto Rosa 24, 4000-098 Porto, Portugal
4   LAETA/INEGI, Faculty of Engineering, University of Porto, Rua Dr. Roberto Frias, 4200-465 Porto, Portugal
*   Correspondence: alexandra.galhano@ulusofona.pt

**Abstract:** A numerical technique was developed for solving nonlocal nonlinear stochastic delayed differential equations driven by fractional variable-order Brownian noise. Error analysis of the proposed technique was performed and discussed. The method was applied to the nonlocal stochastic fluctuations of the human body and the Nicholson's blowfly models, and its accuracy and computational time were assessed for different values of the nonlocal order parameters. A comparison with other techniques available in the literature revealed the effectiveness of the proposed scheme.

**Keywords:** fractional stochastic delayed differential equation; variable-order fractional Brownian noise; cubic spline interpolation; fractional calculus

**MSC:** 26A33; 34K37; 60H35; 65C30; 65D07; 60G22

## 1. Introduction

Fractional calculus (FC) deals with the differentiation and integration operators of arbitrary orders [1–3]. Over the last decades, FC has become globally recognized in nearly all fields due to its widespread usage, such as in mathematics [4–7], mechanics [8–10], physics [11,12], biology [13,14], and economics [15,16]. Fractional differential equations (FDEs) with and without noise are powerful mathematical tools for investigating phenomena in many applied sciences. The FDEs are studied in various branches of science, including chaotic oscillations [17,18], engineering [19,20], image processing [21,22], and epidemic models [23–25]. The existence and uniqueness studies involving FDEs were reviewed in [26,27]. In addition, many papers present distinct numerical techniques for various classes of FDEs, those including Chebyshev polynomials [28–30], finite difference [31], Hermite wavelet [32], Jacobi collocation [33], Legendre collocation [34], and others [35–37]. Recently, a novel exponential time differencing (ETD-RDP) method was developed by Iyiola et al. [38] to handle space-fractional reaction–diffusion systems with mixed and mismatched initial and boundary conditions for which regular numerical methods are unsuitable.

Variable-order fractional operators preserve memory and the hereditary properties of dynamical systems, while the constant-order fractional operators characterize memory with a uniform pattern [39,40]. Fractional differential equations of variable-order include those with classical operators, as well as others with short memory operators, which constitute a new kind of variable-order method [41–43]. Moreover, constant-order fractional operators are a special case (and probably the simplest one) of variable-order fractional ones.

The *variable-order fractional Brownian motion* (VOFBM) is expressed as a Gaussian process via the variable-order Hurst index [44]. The FDEs driven by VOFBM noise have

several usages, such as in physics [45], finance [46], and signal processing [47]. The VOFBM is defined as [44]:

$$\varpi^{H(t)}(t) = \frac{1}{\Gamma(H(t)+\frac{1}{2})} \int_0^t (t-\tau)^{H(t)-\frac{1}{2}} \varpi(\tau)d\tau, \ \frac{1}{2} < H(t) < 1, \qquad t \geq 0. \tag{1}$$

In the follow-up, we assume that $(\Psi, \mathcal{F}, P)$ denotes a fixed probability space with normal filtration $(\mathcal{F}_t)_{t \geq 0}$.

There are distinct kinds of fractional uncertain differential equations [48]. In this paper, we studied the fractional stochastic delay differential equation driven by VOFBM (FSDDE-VOFBM), given by:

$$\begin{cases} {}^C\mathscr{D}_{0,t}^\gamma u(t) = G(t, u(t), u(t-\lambda) + M(t, u(t))\dfrac{d\varpi^{H(t)}(t)}{dt}, & t \in (0, T] \\ u(t) = \Theta(t), & t \in [-\lambda, 0], \end{cases} \tag{2}$$

where $\gamma \in (\frac{1}{2}, 1)$ and ${}^C\mathscr{D}_{0,t}^\gamma u(t)$ denote the Caputo fractional derivative [49]:

$$ {}^C\mathscr{D}_{0,t}^\gamma u(t) = \frac{1}{\Gamma(b-\gamma)} \int_0^t \frac{u^{(b)}(\phi)}{(t-\phi)^{\gamma+1-b}}d\phi, \ 0 \leq b-1 < \gamma \leq b \in \mathbb{N}, \tag{3}$$

where $\gamma \in \mathbb{R}^+$ is the order and $u^{(b)}(t)$ represents a smooth and continuously differentiable function on the interval $\Psi = [0, T]$. Additionally, $G \in \mathbb{C}(\Psi, \mathbb{R}, \mathbb{R})$, $M \in \mathbb{C}(\Psi, \mathbb{R})$, $\lambda$ represents a time delay, $\Theta(t)$ stands for a function on the interval $t \in [-\lambda, 0]$ related to system history, and $\varpi(t)$ $(t \geq 0)$ is a Wiener's process.

There are several definitions of fractional derivative [50,51]. Commonly used formulations are the Grünwald–Letnikov, Riemann–Liouville and Caputo ones. In this paper, we adopted the Caputo fractional derivative. Indeed, it is the most frequently used in engineering and physics applications, since it allows traditional initial and boundary conditions to be included in the corresponding fractional equations, and the Caputo derivative of a constant is zero. Additionally, the Riemann–Liouville definition can be transformed into the Caputo one using an auxiliary power function. The Caputo derivative is an appropriate mean for modeling phenomena characterized by interactions with the past and nonlocal properties.

In this work, a numerical technique was developed for solving the nonlocal nonlinear FSDDE-VOFBM. An error analysis was performed and discussed. The effectiveness of the new method was assessed using some examples, for different values of the nonlocal order parameters. Therefore, the main novelty and contributions of the manuscript can be summarized as follows:

- An accurate and computationally efficient technique for solving FSDDE-VOFBM with Hurst index was proposed;
- A cubic spline interpolation method for time discretization was adopted;
- Error and convergence analysis of the suggested scheme was performed;
- The proposed numerical technique was applied to fractional stochastic dynamical systems and assessed from the perspective of statistical indicators of stochastic responses.

It should be mentioned that the method is valid for nonlocal nonlinear stochastic delayed differential equations using fractional variable-order Brownian noise and, thus, goes beyond other techniques [52,53], which focus on the existence and uniqueness of solutions for different classes of stochastic delay differential equations driven by fractional Brownian motion with the Hurst parameter $H > 1/2$.

The rest of this paper is divided into four sections. Section 2 proposes an explicit method based on cubic spline interpolation to discretize and solve the FSDDE-VOFBM (2). It describes an error analysis of the technique. Section 3 assesses the method accuracy, considering the nonlocal stochastic fluctuation of the human body and the Nicholson's blowfly model. Finally, Section 4 summarizes the main considerations.

## 2. Computational Implementation

An explicit approach for solving the FSDDE-VOFBM (2) was proposed, and error analysis was performed. In the follow-up we considered that $t_l = l\Delta$, with $\Delta = \left[\frac{T}{\varrho}\right]$ denoting an uniform step size and $l = \{0, 1, \dots, \varrho\}$, with $\varrho \in \mathbb{N}$. The cubic spline $s_\varrho(\phi)$ is of the form

$$\frac{\mathrm{d}^b}{\mathrm{d}\phi^b} u(\phi) \approx s_\varrho(\phi) = \sum_{l=0}^{\varrho} N_{l,\varrho}(\phi) \frac{\mathrm{d}^b}{\mathrm{d}\phi^b} u_l + \sum_{l=0}^{\varrho} M_{l,\varrho}(\phi) \frac{\mathrm{d}^{b+1}}{\mathrm{d}\phi^b} u_{l+1}, \tag{4}$$

where the shape functions are stated as $N_{l,\varrho}(\phi)$ and $M_{l,\varrho}(\phi)$ in each interval $[t_l, t_{l+1}]$, for $1 \leq l \leq \varrho - 1$, given by

$$N_{l,\varrho}(\phi) = \begin{cases} \left(1 - \dfrac{2\phi - 2t_l}{t_l - t_{l+1}}\right)\left(\dfrac{\phi - t_{l+1}}{t_l - t_{l+1}}\right)^2, & t_{l-1} \leq \phi \leq t_l \\[2mm] \left(1 - \dfrac{2\phi - 2t_{l+1}}{t_{l+1} - t_l}\right)\left(\dfrac{\phi - t_l}{t_{l+1} - t_l}\right)^2, & t_l \leq \phi \leq t_{l+1} \\[2mm] 0, & \text{otherwise} \end{cases},$$

and

$$M_{l,\varrho}(\phi) = \begin{cases} (\phi - t_l)\left(\dfrac{\phi - t_{l+1}}{t_l - t_{l+1}}\right)^2, & t_{l-1} \leq \phi \leq t_l \\[2mm] (\phi - t_{l+1})\left(\dfrac{\phi - t_l}{t_{l+1} - t_l}\right)^2, & t_l \leq \phi \leq t_{l+1} \\[2mm] 0, & \text{otherwise} \end{cases}.$$

For $l = \{0, \varrho\}$, $N_{l,\varrho}(\phi)$ and $M_{l,\varrho}(\phi)$ are of the form

$$\begin{cases} N_{0,\varrho}(\phi) = \left(1 - \dfrac{2\phi - 2t_1}{t_1 - t_0}\right)\left(\dfrac{\phi - t_0}{t_1 - t_0}\right)^2, & t_0 \leq \phi \leq t_1 \\[2mm] N_{\varrho,\varrho}(\phi) = \left(1 - \dfrac{2\phi - 2t_\varrho}{t_\varrho - t_{\varrho+1}}\right)\left(\dfrac{\phi - t_{\varrho+1}}{t_\varrho - t_{\varrho+1}}\right)^2, & t_{\varrho-1} \leq \phi \leq t_\varrho \\[2mm] N_{0,\varrho}(\phi) = N_{\varrho,\varrho}(\phi) = 0, & \text{otherwise} \end{cases},$$

and

$$\begin{cases} M_{0,\varrho}(\phi) = (\phi - t_1)\left(\dfrac{\phi - t_0}{t_1 - t_0}\right)^2, & t_0 \leq \phi \leq t_1 \\[2mm] M_{\varrho,\varrho}(\phi) = (\phi - t_\varrho)\left(\dfrac{\phi - t_{\varrho+1}}{t_\varrho - t_{\varrho+1}}\right)^2, & t_{\varrho-1} \leq \phi \leq t_\varrho \\[2mm] M_{0,\varrho}(\phi) = M_{\varrho,\varrho}(\phi) = 0, & \text{otherwise} \end{cases}.$$

Therefore, we get

$$\begin{aligned} {}^{C}\mathscr{D}_{0,t_\varrho}^{\gamma}[u(t)] \quad &\approx \quad \left({}^{C}\mathscr{D}_{0,t_\varrho}^{\gamma}[u(t)]\right)_{approx} \\[2mm] &\equiv \quad \sum_{l=0}^{\varrho} \left(\int_{t_l}^{t_{l+1}} \frac{(t_\varrho - \zeta)^{b-1-\gamma}}{\Gamma(b-\gamma)} N_{l,\varrho}(\phi)\mathrm{d}\phi\right) \frac{\mathrm{d}^b}{\mathrm{d}\phi^b} u(t_l) \\[2mm] &+ \quad \sum_{l=0}^{\varrho} \left(\int_{t_l}^{t_{l+1}} \frac{(t_\varrho - \phi)^{b-1-\gamma}}{\Gamma(b-\gamma)} M_{l,\varrho}(\phi)\mathrm{d}\phi\right) \frac{\mathrm{d}^{b+1}}{\mathrm{d}\phi^{b+1}} u(t_l). \end{aligned} \tag{5}$$

and, after some calculations, we obtain

$$\mathstrut^{C}\mathscr{D}_{0,t_\varrho}^{\gamma}[u(t)] \approx \sum_{l=0}^{\varrho} \frac{\Delta^{b-\gamma}}{\Gamma(b-\gamma+4)} \alpha_{l,\varrho} u_l^{(b)} + \sum_{l=0}^{\varrho} \frac{\Delta^{b-\gamma+1}}{\Gamma(b-\gamma+4)} \beta_{l,\varrho} u_l^{(b+1)}, \tag{6}$$

where

$$\alpha_{l,\varrho} = \begin{cases} -6(2\varrho + 1 + b - \gamma)(\varrho - 1)^{b-\gamma+2} + \varrho^{b-\gamma} \times \Big( (-6(b-\gamma) - 18)\varrho^2 \\ \qquad + 12\varrho^3 + (b-\gamma)^3 + 6(b-\gamma)^2 + 11(b-\gamma) + 6 \Big), & l = 0 \\[2ex] 6\Big( (\varrho - l - 1)^{b-\gamma+2}(2l - 2\varrho - b + \gamma - 1) \\ \qquad + (\varrho - l + 1)^{b-\gamma+2}(2l - 2\varrho - b + \gamma + 1 + 4(\varrho - l)^{b-\gamma+3}) \Big), & 1 \le l \le \varrho - 1 \\[2ex] 6(b - \gamma + 1), & l = \varrho \end{cases},$$

and

$$\beta_{l,\varrho} = \begin{cases} -(6\varrho + 2(b-\gamma))(\varrho - 1)^{b-\gamma+2} + n^{b-\gamma+1} \times \\ \quad \Big( b - \gamma^2 + (-4\varrho + 5)(b - \gamma) + 6(\varrho - 1) \Big), & l = 0 \\[2ex] 2(3l - 3\varrho - b + \gamma)(\varrho - l - 1)^{b-\gamma+2} \\ \quad - 2(3l - 3\varrho + b - \gamma)(\varrho - l + 1)^{b-\gamma+2} - 8(\varrho - l)^{b-\gamma+2}(b - \gamma + 3), & 1 \le l \le \varrho - 1 \\[2ex] -2(b - \gamma), & l = \varrho \end{cases}.$$

Thus, we get the following proposition.

**Proposition 1.** *Let us consider the function* $u(t) \in C^{b+4}(\Psi)$, $\gamma > 0$, *and* $\|u^{(b+4)}\|_\infty \le A$, *with* $A > 0$. *Therefore, for* (6), *the truncated error* $\mathcal{R}_\varrho = {}^C\mathscr{D}_{0,t_\varrho}^\gamma[u(t)] - \Big( {}^C\mathscr{D}_{0,t_\varrho}^\gamma[u(t)] \Big)_{approx}$ *is bounded, such that*

$$\mathbb{E}\Big[ |\mathcal{R}_\varrho| \Big] \le \frac{nA}{4! \times 16\Gamma(b + 1 - \gamma)} \Delta^{b-\gamma+4}. \tag{7}$$

**Proof.** Assume the error function, $\mathscr{E}(t)$, defined by

$$\mathscr{E}_l(t) = u_l^{(m)}(t) - s_l(t) = \frac{u^{(b+4)}(\varphi_l)}{4!}(t - t_l)^2(t - t_{l+1})^2, \; l = 1, \dots, \varrho, \tag{8}$$

where $\varphi_l \in [t_l, t_{l+1}]$. Thus,

$$\begin{aligned} \mathbb{E}\Big[ |\mathcal{R}_\varrho| \Big] &= \mathbb{E}\Bigg[ \frac{1}{\Gamma(b-\gamma)} \int_{t_0}^{t_\varrho} (t_\varrho - \phi)^{b-\gamma-1} \|\mathscr{E}(\phi)\|_\infty d\phi \Bigg] \\ &= \frac{1}{\Gamma(b-\gamma)} \mathbb{E}\Bigg[ \sum_{l=0}^{\varrho-1} \int_{t_l}^{t_{l+1}} (t_\varrho - \phi)^{b-\gamma-1} \Big\| \frac{u^{(b+4)}(\varphi_l)}{4!}(\phi - t_l)^2(\phi - t_{l+1})^2 \Big\|_\infty d\phi \Bigg] \\ &\le \frac{Ah^4}{4! \times 16\Gamma(b-\gamma)} \sum_{l=0}^{\varrho-1} \int_{t_l}^{t_{l+1}} (t_\varrho - \phi)^{b-\gamma-1} d\phi \\ &\le B\Delta^{b-\gamma+4}, \end{aligned}$$

where

$$B = \frac{nA}{4! \times 16\Gamma(b + 1 - \gamma)}.$$

$\square$

Hereafter, the proposed algorithm will be briefly denoted as "**CSM**-algorithm". It should be noted that, for solving the initial condition problems such as (2), we incorporated the **CSM**-algorithm with the finite differential quotient stated as:

$$u^{(b)}(t) = \frac{1}{\Delta^b} \sum_{s=0}^{b} (-1)^s \begin{pmatrix} b \\ s \end{pmatrix} u(t + (p - s)\Delta) + \mathcal{O}(\Delta). \tag{9}$$

### 3. Numerical Results and Discussion

We assess the computational effort and accuracy of the proposed method, using the *experimental convergence order* (ECO) and *expected mean absolute error* (EMAE), given by:

$$ECO_{ms} = \log_2 \left( \frac{\|\bar{\mathcal{E}}_{2\varrho}\|_{ms}}{\|\bar{\mathcal{E}}_{\varrho}\|_{ms}} \right), \tag{10}$$

and

$$\|\bar{\mathcal{E}}_{\varrho}\|_{ms} = \frac{1}{\varrho} \sum_{l=1}^{\varrho} \left( \mathbb{E}\left[ \|u_l^{\varrho} - u_{2l}^{2\varrho}\|^2 \right] \right)^{\frac{1}{2}}, \tag{11}$$

where $u_l^{\varrho}$ and $u_{2l}^{2\varrho}$ stand for the approximate values of $u(t_l)$, and $\varrho$ denotes the quantity of mesh interior points. Numerical experiments were performed with the software package Maple V2019 on a processor Intel (R) Core (TM) i7-7500U @ 2.70 GHz.

The approximate solutions and CPU time obtained with the proposed **CSM**-algorithm were compared with the **IQM** [54] and **BSM** methods [55].

**Example 1.** *We considered the fractional stochastic fluctuation of the human body, described as an inverted pendulum under the action of a time-delayed restoring force:*

$$I\,^C\mathscr{D}_{0,t}^{2\gamma}\theta(t) + \phi\,^C\mathscr{D}_{0,t}^{\gamma}\theta(t) - mgl\sin\theta(t) = \tilde{\chi}(\theta(t-\lambda)) + \sqrt{2\widetilde{R}}\frac{d\varpi^{H(t)}(t)}{dt}, \tag{12}$$

*where $^C\mathscr{D}_{0,t}^{2\gamma}u(t)$ denotes the Caputo fractional derivative, with $\gamma \in (\frac{1}{2}, 1)$, and $I = ml^2$ and $mg$ are the moment of inertia and the weight of the pendulum, respectively. Figure 1 illustrates the schematic diagram of this model. Assuming that the sway angle complies with $\theta \ngtr 5°$, then we have*

$$\begin{cases} ^C\mathscr{D}_{0,t}^{\gamma}u(t) = \kappa u(t) + \chi(u(t-\lambda)) + \sigma\frac{d\varpi^{H(t)}(t)}{dt}, & t \in (0, 10] \\ u(t) = 0.1, & t \in [-\lambda, 0], \end{cases} \tag{13}$$

*where $\frac{1}{2} < \gamma, H(t) \leq 1$,*

$$\chi(u(t-\lambda)) = \eta \tanh(u(t-\lambda)),$$

*and $\kappa \approx \sqrt{mgl/2I}$. Moreover, the delay is indicated as $\lambda$, the negative feedback coefficient is introduced as $\eta$, and $\sigma = \sqrt{2\widetilde{R}}$ is used for the value of the noise. Equation (13) was investigated for $H(t) = \frac{1}{2}$ and different values of $\gamma \in (0, 1]$ in [56].*

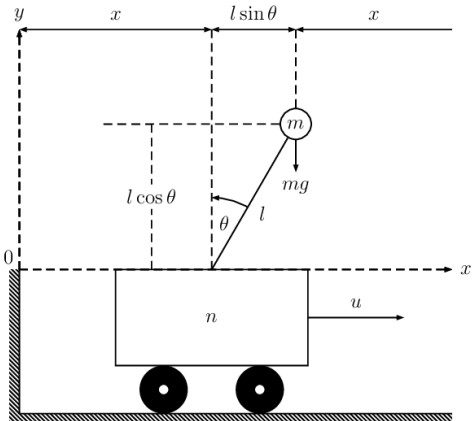

**Figure 1.** Inverted pendulum schematic chart stabilized by means of wain movements.

Figure 2 depicts the approximation of (13) for $\sigma = \{0, 0.01\}$ and $\gamma = \{0.55, 0.75, 0.95\}$, in $t \in [0, 10]$. Moreover, Table 1 compares $\|\bar{\bar{\mathscr{E}}}_{\varrho}\|_{ms}$, $ECO_{ms}$ and computational time obtained with the **IQM-** [54] and **CSM**-algorithms, for distinct values of $\Delta$, $\gamma = \{0.25, 0.5, 0.7, 0.9\}$, and $t \in [0, 10]$. We verified that, for all values of $\gamma$, the errors yielded by the proposed method decrease as $\Delta$ diminishes. Table 2 lists the values of some statistical indicators (SIs) for several fractional orders, with $T = 10$. We verified that the values of the median and mean are equal for the fractional orders $\gamma = \{0.55, 0.75, 0.95\}$. This means that the diagram driven by 50 simulated paths at $T = 10$ is symmetric.

**Table 1.** Example 1: The values of $\|\bar{\bar{\mathscr{E}}}_{\varrho}\|_{ms}$, $ECO_{ms}$, and CPU time (expressed in seconds) for (13) obtained with the **IQM-** [54] and **CSM**-algorithms for distinct choices of $\gamma$ and $\Delta$, with $\kappa = 1.58$, $\eta = -1.6$, $\sigma = 0.01$, $\lambda = 0.1$, and $H(t) = 0.95 - 0.02\,t$, in $t \in [0, 10]$.

| | | IQM-Algorithm [54] | | | CSM-Algorithm | | |
|---|---|---|---|---|---|---|---|
| $\gamma$ | $\Delta$ | $\|\bar{\bar{\mathscr{E}}}_{\varrho}\|_{ms}$ | $ECO_{ms}$ | *CPU Time* | $\|\bar{\bar{\mathscr{E}}}_{\varrho}\|_{ms}$ | $ECO_{ms}$ | *CPU Time* |
| | 0.02 | $2.18 \times 10^{-4}$ | – | 35.703 | $7.31 \times 10^{-5}$ | – | 26.344 |
| 0.55 | 0.01 | $1.12 \times 10^{-4}$ | 0.96 | 151.516 | $6.31 \times 10^{-6}$ | 3.53 | 114.860 |
| | 0.005 | $6.04 \times 10^{-5}$ | 0.89 | 708.140 | $1.13 \times 10^{-6}$ | 2.48 | 494.953 |
| | 0.02 | $2.49 \times 10^{-4}$ | – | 35.578 | $1.47 \times 10^{-4}$ | – | 25.719 |
| 0.75 | 0.01 | $1.36 \times 10^{-4}$ | 0.87 | 147.922 | $1.01 \times 10^{-5}$ | 3.86 | 111.859 |
| | 0.005 | $7.39 \times 10^{-5}$ | 0.88 | 704.922 | $5.02 \times 10^{-6}$ | 2.32 | 490.843 |
| | 0.02 | $5.65 \times 10^{-4}$ | – | 35.641 | $2.84 \times 10^{-4}$ | – | 25.438 |
| 0.95 | 0.01 | $3.21 \times 10^{-4}$ | 0.85 | 151.937 | $1.60 \times 10^{-5}$ | 4.13 | 109.656 |
| | 0.005 | $1.57 \times 10^{-4}$ | 1.00 | 705.917 | $4.04 \times 10^{-6}$ | 1.98 | 499.797 |

**Table 2.** The approximated SI values concerning the 50 simulated paths for (13), with $\gamma = \{0.55, 0.75, 0.95\}$, $\kappa = 1.58$, $\eta = -1.6$, $\lambda = 0.1$, $\sigma = 0.01$, $H(t) = 0.95 - 0.02\,t$, and step size $\Delta = 0.01$, at $T = 10$.

| SI | $\gamma = 0.55$ | $\gamma = 0.75$ | $\gamma = 0.95$ |
|---|---|---|---|
| Mean | 0.093 | 0.089 | 0.083 |
| Median | 0.093 | 0.089 | 0.083 |
| First quartile | 0.091 | 0.088 | 0.081 |
| Third quartile | 0.094 | 0.092 | 0.087 |
| Kurtosis | 2.350 | 2.582 | 2.944 |
| Skewness | 0.200 | $-0.038$ | $-0.197$ |
| Standard deviation | $2.268 \times 10^{-3}$ | $3.661 \times 10^{-3}$ | $5.975 \times 10^{-3}$ |
| 95% Confidence interval | $[0.089, 0.097]$ | $[0.083, 0.095]$ | $[0.073, 0.093]$ |

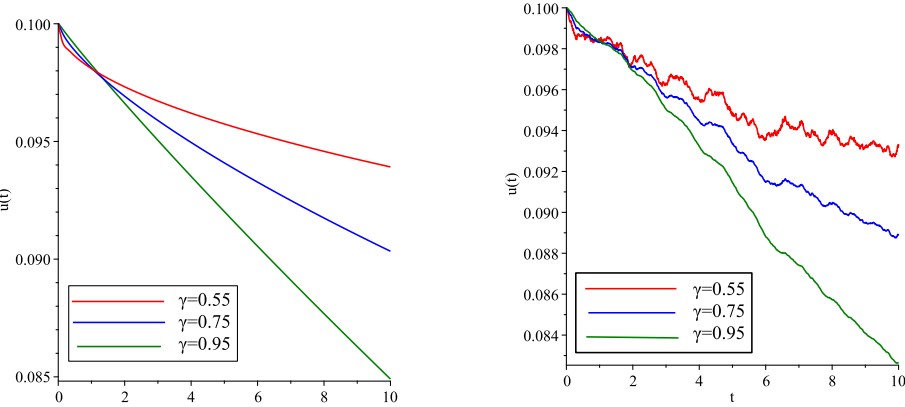

**Figure 2.** The time evolution of $u(t)$ for (12) with the proposed algorithm for $\kappa = 1.58$, $\eta = -1.6$, $\lambda = 0.1$, $\gamma = \{0.55, 0.75, 0.95\}$, $H(t) = 0.95 - 0.02t$, and $\Delta = 0.01$: (**left side**) $\sigma = 0$; (**right side**) $\sigma = 0.01$.

**Example 2.** *We considered the nonlinear fractional stochastic Nicholson's blowflies differential equation with time delay:*

$$
\begin{cases}
{}^{C}\mathscr{D}_{0,t}^{\gamma} u(t) = \kappa u(t - \lambda) \exp(-\mu u(t - \lambda)) \\
\qquad\qquad - \rho u(t) + \sigma \left( u(t) - \mu^{-1} \ln\left( \kappa \rho^{-1} \right) \right) \dfrac{d\omega^{H(t)}(t)}{dt}, & t \in [0, 10] \\
u(t) = 1.35 \cos(3t), & t \in [-\lambda, 0]
\end{cases}
\tag{14}
$$

*where $\frac{1}{2} < \gamma, H(t) \leq 1$, $u(t)$ indicates the crowd size at the time instant $t$, $\kappa$ represents peak per capita daily rate of egg production, $\lambda$ denotes production time, $\mu^{-1}$ is the value at which the crowd multiplies at the peak rate, $\rho$ is the adulthood per capita daily death rate, and $\sigma$ introduces the publication coefficient. It should be noted that the model (14), including non-stochastic and stochastic terms, was studied in [57,58].*

Figure 3 illustrates the approximated solutions of (14) for $\kappa = 9$, $\mu = 1$, $\lambda = 0.05$, $\rho = 2$, $H(t) = 0.6 + 0.2 \exp(0.01t)$ and $\sigma = \{0, 5\}$ with different values of $\gamma$ and step size $\Delta = 0.01$. We can observe the noise effect in Equation (14). Moreover, Figure 4 plots the approximation magnitudes of SIs of the 50 simulated paths for $\sigma = 5$, $\gamma = 0.55$ at $T = 10$.

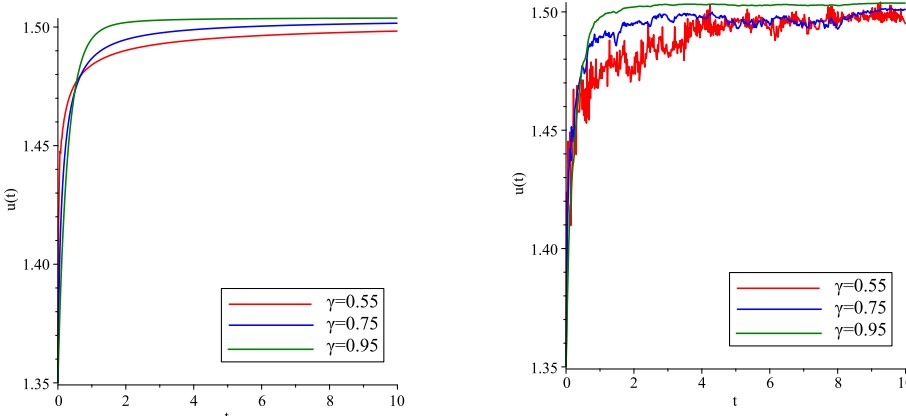

**Figure 3.** The time evolution of $u(t)$ for (14) with the proposed algorithm for $\kappa = 9$, $\mu = 1$, $\lambda = 0.05$, $\rho = 2$, $\gamma = \{0.55, 0.75, 0.95\}$, $H(t) = 0.6 + 0.2 \exp(0.01t)$, and $\Delta = 0.01$: (**left side**) $\sigma = 0$; (**right side**) $\sigma = 5$.

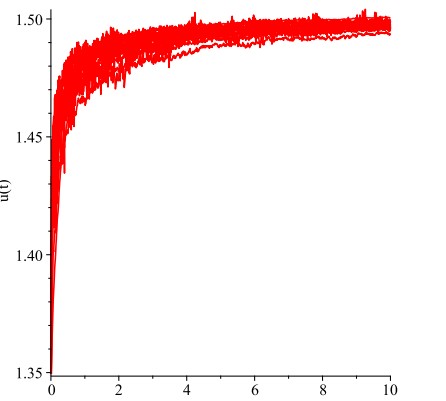 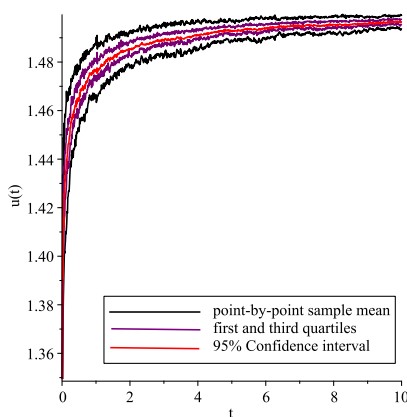

**Figure 4.** The time evolution of $u(t)$ (**left side**) and the SI values (**right side**) for (14) with the proposed algorithm, with $\kappa = 9$, $\mu = 1$, $\lambda = 0.05$, $\rho = 2$, $\sigma = 5$, $\gamma = 0.55$, $H(t) = 0.6 + 0.2 \exp(0.01t)$ and $\Delta = 0.01$ over 50 simulated paths.

Table 3 summarizes $\|\bar{\mathscr{E}}_\varrho\|_{ms}$, $ECO_{ms}$ and the computational time values of (14), obtained with the **BSM-** [55] and **CSM**-algorithms for $\gamma = \{0.55, 0.75, 0.95\}$, with $\Delta = \{0.02, 0.01, 0.005\}$ and $\lambda = 0.05$ in the time interval $t \in [0, 10]$. We verified that a more accurate approximation is obtained when reducing the step size. Table 4 presents the SI values for several $\gamma$ cases, at $T = 10$. We verified that the diagram driven by 50 paths for $\gamma = \{0.55, 0.75, 0.95\}$ at $T = 10$ is symmetric. It should be noted that adopting $\sigma = 5$ corresponds to having data polluted by large random noise. Still, the stability of the proposed method was verified.

Finite difference methods are intuitive and easy to implement. However, the discretization schemes based on finite difference quotients do not necessarily increase convergence order by increasing the number of mesh points, just diminishing the error of the approximation. This limitation is mitigated by our method, which uses a discretization scheme based on cubic spline interpolation, with an implementation complexity similar to that of the finite difference method. The experimental convergence order of the proposed method is listed in Tables 1 and 3.

**Table 3.** Example 2: The values of $\|\bar{\mathscr{E}}_\varrho\|_{ms}$, $ECO_{ms}$ and CPU time (expressed in seconds) for (14) obtained with the **BSM-** [55] and **CSM**-algorithms for different values of $\gamma$ and $\Delta$, with $\kappa = 9$, $\mu = 1$, $\lambda = 0.05$, $\rho = 2$, $\sigma = 5$, and $H(t) = 0.6 + 0.2 \exp(0.01t)$, in $t \in [0, 10]$.

| | | **BSM-Algorithm [55]** | | | **CSM-Algorithm** | | |
|---|---|---|---|---|---|---|---|
| $\gamma$ | $\Delta$ | $\|\bar{\mathscr{E}}_\varrho\|_{ms}$ | $ECO_{ms}$ | *CPU Time* | $\|\bar{\mathscr{E}}_\varrho\|_{ms}$ | $ECO_{ms}$ | *CPU Time* |
| | 0.02 | $1.58 \times 10^{-3}$ | — | 10.140 | $9.54 \times 10^{-4}$ | — | 25.907 |
| 0.55 | 0.01 | $9.88 \times 10^{-4}$ | 0.68 | 44.828 | $4.71 \times 10^{-4}$ | 1.02 | 114.094 |
| | 0.005 | $7.28 \times 10^{-4}$ | 0.44 | 194.265 | $8.68 \times 10^{-5}$ | 2.44 | 495.860 |
| | 0.02 | $5.80 \times 10^{-4}$ | — | 9.937 | $3.98 \times 10^{-4}$ | — | 25.719 |
| 0.75 | 0.01 | $2.91 \times 10^{-4}$ | 0.99 | 4.328 | $1.85 \times 10^{-4}$ | 1.10 | 115.375 |
| | 0.005 | $2.07 \times 10^{-4}$ | 0.49 | 191.828 | $5.43 \times 10^{-5}$ | 1.77 | 508.531 |
| | 0.02 | $4.05 \times 10^{-4}$ | — | 10.687 | $3.96 \times 10^{-4}$ | — | 26.391 |
| 0.95 | 0.01 | $2.26 \times 10^{-4}$ | 0.84 | 45.094 | $1.87 \times 10^{-4}$ | 1.08 | 120.328 |
| | 0.005 | $9.42 \times 10^{-5}$ | 1.26 | 185.047 | $5.89 \times 10^{-5}$ | 1.66 | 494.093 |

**Table 4.** The approximated SI values concerning the 50 simulated paths for (14), with $\gamma = \{0.55, 0.75, 0.95\}$, $\kappa = 9$, $\mu = 1$, $\lambda = 0.05$, $\rho = 2$, $\sigma = 5$, $H(t) = 0.6 + 0.2\exp(0.01t)$ and step size $\Delta = 0.01$, at $T = 10$.

| *SI* | $\gamma = 0.55$ | $\gamma = 0.75$ | $\gamma = 0.95$ |
|---|---|---|---|
| Mean | 1.496 | 1.501 | 1.503 |
| Median | 1.496 | 1.501 | 1.503 |
| First quartile | 1.495 | 1.500 | 1.503 |
| Third quartile | 1.497 | 1.502 | 1.504 |
| Kurtosis | 3.145 | 2.670 | 2.567 |
| Skewness | $-0.834$ | $-0.067$ | 0.215 |
| Standard deviation | $1.490 \times 10^{-3}$ | $4.645 \times 10^{-4}$ | $5.198 \times 10^{-5}$ |
| 95% Confidence interval | $[1.493, 1.498]$ | $[1.500, 1.502]$ | $[1.503, 1.504]$ |

## 4. Conclusions

An explicit scheme based on cubic spline interpolation was proposed for numerically solving nonlocal FSDDE-VOFBM. The effectiveness of the method when applied to the nonlocal stochastic fluctuation of the human body and the Nicholson's blowfly models was investigated. Numerical experiments indicated the performance of the proposed method both in terms of accuracy and computational burden. The results also revealed the efficiency and feasibility of the algorithm for nonlinear stochastic delay systems. In future research, we will consider this technique for tackling models with distributed order fractional derivatives.

**Author Contributions:** Investigation, Methodology and Writing—original draft, B.P.M.; Software and Conceptualization, M.P. and Z.S.M.; Writing—review and editing, O.S.I.; visualization and supervision, A.G. and A.M.L. All authors have read and agreed to the published version of the manuscript.

**Funding:** This research received no external funding.

**Data Availability Statement:** Data sharing is not applicable to this article, as no datasets were generated or analyzed during the current study.

**Conflicts of Interest:** The authors declare no conflicts of interest.

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
