# Peer review of "A Numerical Algorithm for Solving Nonlocal Nonlinear Stochastic Delayed Systems with Variable-Order Fractional Brownian Noise"

_fractalfract, doi:10.3390/fractalfract7040293_

Round 1

Reviewer 1 Report

In this research, a numerical technique is developed for solving nonlocal nonlinear stochastic delayed differential equations with fractional variable-order Brownian noise. Authors have studied the error analysis of the proposed technique.

I found little novelty of the research. Authors are requested to highlight the novelty of their research. Also mention how your manuscript differ from the following articles:

1. 10.1186/s13662-017-1210-6

2. 10.1016/j.camwa.2009.05.004

Author Response

Reviewer 1

In this research, a numerical technique is developed for solving nonlocal nonlinear stochastic delayed differential equations with fractional variable-order Brownian noise. Authors have studied the error analysis of the proposed technique.

I found little novelty of the research. Authors are requested to highlight the novelty of their research. Also mention how your manuscript differ from the following articles:

1. 10.1186/s13662-017-1210-6
2. 10.1016/j.camwa.2009.05.004

Response to Reviewer 1:
Thank you for your suggestion. The motivation and the novelty of the paper are highlighted in the revised version of the manuscript. Regarding novelty of the manuscript, we believe that it contains some valuable new information. Indeed, it is well known that variable-order fractional operators preserve the memory and hereditary properties of dynamical systems, while the constant-order fractional operators characterize the memory of dynamical systems with a uniform pattern. Moreover, constant-order fractional operators are a special case (and probably the simplest one) of variable-order fractional operators. The proposed method can be also used for solving linear and nonlinear delay stochastic differential equations with constant-order fractional operators.

Therefore, the main novelty of the manuscript can be summarized as follows:

  • An accurate and computationally efficient technique for solving fractional stochastic differential equations driven by fractional Brownian motion with Hurst index is proposed.
  • A cubic spline interpolation method for time discretization is adopted.
  • Error and convergence analysis of the suggested scheme is performed.
  • The proposed numerical technique is applied to fractional stochastic dynamical systems, and assessed in the perspective of statistical indicators of stochastic responses.

The significant differences between [1,2] and our study is that in [1,2] the authors concentrate on the existence and uniqueness of solutions for different classes of stochastic delay differential equations driven by fractional Brownian motion with the Hurst parameter H>1/2. But we propose a numerical technique for nonlocal nonlinear stochastic delayed differential equations using fractional variable-order Brownian noise.

Reviewer 2 Report

The considered paper contains new and interested results

Author Response

Reviewer 2
The considered paper contains new and interested results.

Response to Reviewer 2:
We would like to thank you for your positive opinion.

Reviewer 3 Report

Moreover, some drawbacks are still not be addressed:

(1) Future recommendations should be added to assist other researchers to extend the presented research analysis.

(2) In the introduction, the authors did not provide a strong motivation for the paper and the obtained results. In addition, they should discuss the main contributions of their work in detail after the motivation part. Then, they should summarize the main structure of his paper in brief at the end of the introduction.

(3) The English writing of the paper is required to be improved. Please check the manuscript carefully for typos and grammatical errors. The reviewer found some typos and grammatical errors within this manuscript, which have been excluded from my review. In addition, the English structure of the article, including punctuation, semicolon, and other structures, must be carefully reviewed.

(4) The authors have to compare their results with the other literature and present the accuracy.

(5) The reviewer interests in the stability of the proposed method when the input data are polluted by large random noise for different issues. However, the results do not demonstrate in the manuscript.

(6) The reviewer concerns about the error analysis of the proposed method in the revision.

(7) In the manuscript, the author does not show the order of convergence of his methods. What's the order of convergence?

Author Response

Reviewer 3

Moreover, some drawbacks are still not be addressed:

Future recommendations should be added to assist other researchers to extend the presented research analysis.

Response to Reviewer 3:

This is an interesting suggestion. For future research we will consider this technique for tackling models with distributed order fractional derivatives.

Reviewer 3

In the introduction, the authors did not provide a strong motivation for the paper and the obtained results. In addition, they should discuss the main contributions of their work in detail after the motivation part. Then, they should summarize the main structure of his paper in brief at the end of the introduction.

Response to Reviewer 3:
Thank you for your suggestion. The motivation and the novelty of the paper are highlighted in the revised version of the manuscript. Regarding novelty of the manuscript, we believe that it contains some valuable new information. Indeed, it is well known that variable-order fractional operators preserve the memory and hereditary properties of dynamical systems, while the constant-order fractional operators characterize the memory of dynamical systems with a uniform pattern. Moreover, constant-order fractional operators are a special case (and probably the simplest one) of variable-order fractional operators. The proposed method can be also used for solving linear and nonlinear delay stochastic differential equations with constant-order fractional operators.

Therefore, the main novelty of the manuscript can be summarized as follows:

  • An accurate and computationally efficient technique for solving fractional stochastic differential equations driven by fractional Brownian motion with Hurst index is proposed.
  • A cubic spline interpolation method for time discretization is adopted.
  • Error and convergence analysis of the suggested scheme is performed.
  • The proposed numerical technique is applied to fractional stochastic dynamical systems, and assessed in the perspective of statistical indicators of stochastic responses.

.Reviewer 3

The English writing of the paper is required to be improved. Please check the manuscript carefully for typos and grammatical errors. The reviewer found some typos and grammatical errors within this manuscript, which have been excluded from my review. In addition, the English structure of the article, including punctuation, semicolon, and other structures, must be carefully reviewed.

Response to Reviewer 3:
Done. All typos and corrections have been made in the revised version.

Reviewer 3

The authors have to compare their results with the other literature and present the accuracy.

Response to Reviewer 3:
Thank you for your constructive comments. We compare the results obtained with our method with others found in the literature. Please consider tables 1 and 3.

Reviewer 3

The reviewer interests in the stability of the proposed method when the input data are polluted by large random noise for different issues. However, the results do not demonstrate in the manuscript.

Response to Reviewer 3:
Thanks. We studied a fractional stochastic delay differential equation with $\sigma=5$, by large random noise. Please consider Example 2.

Reviewer 3

The reviewer concerns about the error analysis of the proposed method in the revision.

Response to Reviewer 3:
The error analysis was presented in Proposition 1.

Reviewer 3

In the manuscript, the author does not show the order of convergence of his methods. What's the order of convergence?

Response to Reviewer 3:
Finite difference methods are intuitive and easy to implement. However, the discretization schemes based on finite difference quotients do not necessarily increase convergence order by increasing the number of mesh points, just diminishing the error of the approximation. This limitation is mitigated by our method, which uses a discretization scheme based on cubic spline interpolation, with implementation complexity similar to that of the finite difference method. The experimental convergence order of proposed method is listed in the Tables 1 and 3.

Author Response

Reviewer 4

This paper proposes a numerical technique for solving nonlocal nonlinear stochastic delayed differential equations with fractional variable-order Brownian noise. This topic is interesting, but needs the following revision:
As we all know, there are many definitions of the fractional derivative such as local fractional derivative, Conformable fractional derivative and so on. Why the Caputo fractional derivative is used? Please give the physical meaning of the Caputo fractional derivative in this work.

Response to Reviewer 4:
Thank you for your positive opinion and constructive comments. There are several definitions of fractional derivative. Some of them introduced in the following reference:

De Oliveira, Edmundo Capelas, and J. A. Tenreiro Machado. A Review of Definitions for Fractional Derivatives and Integral." Mathematical Problems in Engineering 2014 (2014): 1-6. doi:10.1155/2014/238459

Commonly used formulations are the Grunwald-Letnikov, Riemann-Liouville  and Caputo ones. In this paper, we adopt the Caputo fractional derivative. Indeed, it is the most frequently used in engineering and physics applications, since it allows traditional initial and boundary conditions to be included in the corresponding fractional differential equations, and the derivative of a constant is zero. Besides, the Riemann-Liouville definition can be transformed into the Caputo one using an auxiliary power function. It should be noted that some, as the conformable derivative, are not fractional derivatives. For more information, please consider following reference:

Valério, Duarte, Manuel D. Ortigueira, and António M. Lopes. "How many fractional derivatives are there?." Mathematics 10, no. 5 (2022): 737.

The Caputo derivative is an appropriate mean for modeling phenomena characterized by interactions with the past and nonlocal properties.

Reviewer 4

Add the main finding and objective of the current study in the abstract.

 Response to Reviewer 4:

Thank you for your positive opinion and constructive comments. Abstract is improved in the revised paper.

Reviewer 4

The convergence of the proposed method should be discussed in detail.

Response to Reviewer 4:
The error analysis and error upper bounds are presented in Proposition 1.

Reviewer 4

English writing should be improved much.

Response to Reviewer 4:
Thanks, we tried to correct the typos and grammatical errors presented in the previous version of this manuscript.

Reviewer 4

The recent results on the fractional derivative should be discussed:

Fractal and Fractional, 2023, 7 (1), 72; Results Phys, 10, 2018 272-276.

Response to Reviewer 4:
Thank you. We included all valuable references.

Reviewer 4

Please check the all equations to make sure they are correct.

Response to Reviewer 4:

Thanks for pointing out that detail. All equations have been checked.

Reviewer 5 Report

The authors proposed an explicit scheme for the numerical solution of nonlocal delay differential equations driven by a variable-order fractional Brownian motion, based on cubic spline interpolation. The results are derived efficiently and correctly. Some numerical results are proposed to support the theoretical derivations.  

The paper has only one drawback concerning with the stability of the constructed numerical scheme. I would advice the authors to refer to the recent related references that can be helpful in the stability analysis.

Author Response

Comment:

The authors proposed an explicit scheme for the numerical solution of nonlocal delay differential equations driven by a variable-order fractional Brownian motion, based on cubic spline interpolation. The results are derived efficiently and correctly. Some numerical results are proposed to support the theoretical derivations.  

The paper has only one drawback concerning with the stability of the constructed numerical scheme. I would advise the authors to refer to the recent related references that can be helpful in the stability analysis.

Response to Reviewer 5:
Thank you for your positive opinion and constructive comments. Error and convergence analysis of the suggested scheme is investigated. We also studied a fractional stochastic delay differential equation with $\sigma=5$, with large random noise. Please consider Example 2.

As far as we are concerned, the stability of the proposed iterative method is verified.

Reviewer 6 Report

I found the following problems to improve:

(1) Definition 1 has no integral function. 

(2) We generally use n-1<gamma<=n. Why use b?

(3) The authors claimed "u(t) is a (b − 1)-times continuously differentiable " But the function in (3) appears "u^(b)(t)" That is u(t) is a b-times continuously differentiable. So Please explain the function space. 

(4) There is another kind of fractional uncertain differential equation:

G.C. Wu, J.L. Wei, C. Luo, L.L. Huang, Parameter estimation of fractional uncertain differential equations via Adams method, Nonlinear Anal: Model. Contr. 27 (2022) 413–427.

(5) There are short memory fractional differential equations with variable order which is a new kind of variable order method. Please check the related works. 

(6) In Eq. (12), please define: D^2{2gamma}

(7) In Proposition 1, u in C^{b+4}. The function space is too strong. So this condition is too strict and reduces the novelty. 

(8) How about the general fractional calculus which includes Caputo, Hadamard and exponential derivatives? Please mention the fractional differential equations if they are defined by the general fractional calculus. 

Author Response

Comment:

(1) Definition 1 has no integral function. 

Response to Reviewer 6:
Thanks for pointing out that detail. We corrected the issue.

Comment:

(2) We generally use n-1<gamma<=n. Why use b?

Response to Reviewer 6:
We agree. We did this only to reduce the similarity with definitions used in many articles.

Comment:

(3) The authors claimed "u(t) is a (b − 1)-times continuously differentiable " But the function in (3) appears "u^(b)(t)" That is u(t) is a b-times continuously differentiable. So, please explain the function space. 

Response to Reviewer 6:
Thanks for pointing that detail. It has been checked and modified.

Comment:

(4) There is another kind of fractional uncertain differential equation:

  • C. Wu, J.L. Wei, C. Luo, L.L. Huang, Parameter estimation of fractional uncertain differential equations via Adam’s method, Nonlinear Anal: Model. Contr. 27 (2022) 413–427.

Response to Reviewer 6:
Thank you for pointing this paper. We added it and the list of references was updated with latest relevant developments and publications.

(5) There are short memory fractional differential equations with variable order which is a new kind of variable order method. Please check the related works.

Response to Reviewer 6:
Thank you. We mentioned this in the revised paper and added a few references.

Comment:

(6) In Eq. (12), please define: D^2{2gamma}

Response to Reviewer 6:
Thanks for pointing that detail. It has been checked. $^C{\mathscr D}^{2\gamma}_{0,t}u(t)$ denotes the Caputo fractional derivative where  $\gamma\in(\frac{1}{2},1)$,

Comment:

(7) In Proposition 1, u in C^{b+4}. The function space is too strong. So, this condition is too strict and reduces the novelty. 

Response to Reviewer 6:
We agree. Indeed, finite difference methods are intuitive and easy to implement. However, the discretization schemes based on finite difference quotients do not necessarily increase convergence order by increasing the number of mesh points, just diminishing the error of the approximation. This limitation is mitigated by our method, which uses a discretization scheme based on cubic spline interpolation, with implementation complexity similar to that of the finite difference method. The experimental convergence order of the proposed method is listed in Tables 1 and 3.

Comment:

How about the general fractional calculus which includes Caputo, Hadamard and exponential derivatives? Please mention the fractional differential equations if they are defined by the general fractional calculus. 

Response to Reviewer 6:
We believe that the time-delay estimation for non-linear systems is one of the most important topics in oscillation theory of functional stochastic differential equations, which have not been touched by scholars for VOSFDEs. However, the paper already includes several contributions. Adding a new topic to the paper may cause difficulty and confusion to readers. We believe that this can be a topic of a future work and it is out of scope of this paper.

Round 2

Reviewer 1 Report

Authors have revised the manuscript according to the comments and suggestions.

Therefore I accept the manuscript in the present form.

Reviewer 3 Report

The revision can be accepted.

Reviewer 4 Report

The revised paper has been greatly improved and is recommended to be accepted for publication!

Reviewer 6 Report

I have no other further comments